# Fluorescence Microscopy of the HIV-1 Envelope

**DOI:** 10.3390/v12030348

**Published:** 2020-03-21

**Authors:** Pablo Carravilla, José L. Nieva, Christian Eggeling

**Affiliations:** 1Institute of Applied Optics and Biophysics, Friedrich-Schiller-University Jena, Max-Wien Platz 1, 07743 Jena, Germany; christian.eggeling@uni-jena.de; 2Leibniz Institute of Photonic Technology, Albert Einstein Strasse 9, 07743 Jena, Germany; 3Department of Biochemistry and Molecular Biology, University of the Basque Country (UPV/EHU), P.O. Box 644, 48080 Bilbao, Spain; joseluis.nieva@ehu.eus; 4Instituto Biofisika (UPV/EHU, CSIC), University of the Basque Country, E-48940 Leioa, Spain; 5MRC Weatherall Institute of Molecular Medicine, University of Oxford, Headley Way, Oxford OX3 9DS, UK; 6Jena Center for Soft Matter (JCSM), Friedrich-Schiller-University Jena, Philosophenweg 7, 07743 Jena, Germany

**Keywords:** microscopy, HIV, envelope, lipid, membrane, Env, fluorescence, super-resolution

## Abstract

Human immunodeficiency virus (HIV) infection constitutes a major health and social issue worldwide. HIV infects cells by fusing its envelope with the target cell plasma membrane. This process is mediated by the viral Env glycoprotein and depends on the envelope lipid composition. Fluorescent microscopy has been employed to investigate the envelope properties, and the processes of viral assembly and fusion, but the application of this technique to the study of HIV is still limited by a number of factors, such as the small size of HIV virions or the difficulty to label the envelope components. Here, we review fluorescence imaging studies of the envelope lipids and proteins, focusing on labelling strategies and model systems.

## 1. Introduction

Retroviridae family members are responsible for a high number of human diseases and conditions, and also powerful tools for gene delivery. A common feature to retroviruses is the viral envelope. It consists of a lipid bilayer that surrounds the capsid and contains the viral Env glycoprotein. The envelope components are typically acquired from the host cell plasma membrane upon viral budding. Its main functions are to define viral tropism, provide virions with a mechanism to enter target cells, and protect the viral capsid and RNA.

One of the most prominent members of the Retroviridae virus family is the Human Immunodeficiency Virus (HIV). HIV infection causes the acquired immune deficiency syndrome and is responsible for around one million deaths worldwide every year. In the case of the HIV, the Env glycoprotein is formed by the surface gp120 and transmembrane gp41 proteins. Gp120 binds to the CD4 receptor and CCR5/CXCR4 co-receptor, typically found on the membrane of T helper cells of our immune system. After receptor engagement, gp41 catalyses the fusion reaction, merging the viral lipid bilayer with the cell plasma membrane and releasing the capsid to the cytosol. Being the only viral protein exposed on virions, Env is a prime target for the immune system, although it typically fails to raise a long-lasting neutralising response during infection due to the number of immune escape mechanisms put in action by the virus. In some rare cases, the immune system is able to produce broadly neutralising antibodies [1], i.e., immunoglobulins that are able to efficiently bind to Env and block cell entry of a great variety of HIV isolates. Thus, Env has been the subject of intensive research and vaccination efforts.

Envelope lipids also present a number of unique features. Lipidomic studies provided clear evidence that the lipid composition of the viral envelope differs from that of the plasma membrane of producer cells [2,3,4,5]. The enrichment in sphingomyelin and cholesterol suggests that HIV buds from specialised nanodomains in the plasma membrane, as previously evidenced [6] and further confirmed [7,8]. In particular, cholesterol proved to be a critical lipid, since its depletion from virions inhibits HIV infectivity [9,10,11]. However, the exact inhibition mechanism is not clear yet, as viral cholesterol depletion has been reported to alter many viral features, such as membrane integrity [12], molecular order [13,14], phase properties [14] and Env stability [15]. These results suggested that the HIV membrane constitutes a functional viral component during entry. In line with this hypothesis, different membrane-targeting compounds present antiviral activity, as reviewed in [16] and [17]. Still, many open questions remain. The details of Env-mediated viral entry need to be elucidated, especially regarding the interplay between the glycoprotein and the viral membrane, and the role of the latter.

This review is focused on the application of fluorescence microscopy to the study of the HIV envelope and its role during entry and budding. Its application to virology is mainly hindered by three limitations. First, HIV virions are ca. 120 nm in diameter, which is a factor of two below the resolution limit of a conventional light microscope. This does not make HIV virions undetectable, but instead, they appear as 250 nm particles and no detail within them can be resolved, hampering their study. Second, viral entry and budding are minute-lived processes. The observation of such long phenomena often give rise to photobleaching, i.e., destruction of fluorophores, and phototoxic effects. Third, HIV virions are not fluorescent, and the introduction of fluorescence dyes can potentially alter their behaviour. Recent advances in quantitative and super-resolution microscopy, as well as design of novel fluorescent dyes will contribute to overcome these limitations, and some have already been successfully exploited in the HIV field.

## 2. Imaging HIV Membrane Lipids

Two types of fluorescent molecules are typically used to visualise membranes: lipid analogues and membrane dyes (Figure 1). Lipid analogues are lipids conjugated to a fluorescent dye that ideally hardly affects their usual behaviour. Analogues are widely used to image specific lipid species, but they can also be used as simple membrane stains [18]. Membrane dyes are lipophilic fluorophores that partition to lipid bilayers and permit their detection. Some of them also report on the biophysical properties of the membrane, e.g., lipid packing [19] or tension [20]. Generally, lipid analogues have been used to study viral budding and membrane dyes to investigate entry and the properties of the viral membrane, as outlined in the next paragraphs.

### 2.1. Imaging Lipids During HIV Assembly and Budding

The application of advanced optical microscopy to the study of HIV assembly was recently reviewed in this journal [21] and elsewhere [22,23], and instead, we will focus on the study of lipids during the process. HIV budding starts with the assembly of the structural polyprotein Gag at the inner leaflet of the plasma membrane of infected cells. The matrix (MA) domain binds to the plasma membrane inner leaflet, where it forms multimers and generates curvature, ultimately promoting virion release [24,25,26,27]. MA binds the plasma membrane through a phosphatidylinositol (4,5) biphosphate (PIP_2_) binding pocket and myristylation [27,28]. However, whether the unique lipid and protein composition of HIV virions arises from MA binding to pre-existing membrane domains or is actively promoted by Gag has been a matter of debate. 

The existence of specialised membrane environments is widely accepted, but their nature is still under discussion, especially regarding their size and lifetime. The lipid raft hypothesis proposed that ordered membrane platforms enriched in sphingomyelin, cholesterol, and saturated lipids would create chemically distinct membrane environments denoted “lipid rafts”. For a detailed review on membrane organisation, refer to manuscripts such as those by Sezgin et al. [29] or Levental et al. [30]. It was tempting to speculate that HIV acquired its unique membrane by budding from rafts [2], further supported by the punctuate organisation of Gag at the plasma membrane [31]. However, lipid rafts have not been detected on the membrane of live untreated cells [32].

First imaging studies of Gag-lipid interactions were performed in model membranes [33,34,35,36], which constitute useful systems to address protein-lipid interactions due to their tuneable lipid composition. Gag-membrane interactions and subsequent ESCRT machinery recruitment were successfully reconstituted in giant unilamellar vesicles (GUVs) [33]. Further microscopy experiments on GUVs showed that MA binding to lipids depends on myristoylation, PIP_2_ and Gag multimerization [34]. Unexpectedly, MA preferentially bound to disordered “non-raft” environments, where it colocalised with different fluorescent PIP_2_ analogues (labelled with BODIPY TMR (BT) [34] or Top-Fluor (TF) [35]). Combined to a preferential binding to unsaturated lipid chains [36,37], Gag assembly at pre-existing lipid rafts appears unlikely. Supporting the idea of Gag actively promoting budding domains, molecular dynamics experiments using Förster resonance energy transfer (FRET) and fluorescence recovery after photobleaching (FRAP) showed that Gag induced the reorganisation of the fluorescent analogues of cholesterol (labelled with TF) (Figure 1), PIP_2_ (labelled with TF or BT), but not of SM (labelled with TF) [35].

Recent microscopy studies on live cells confirmed that Gag recruits proteins and lipids at budding platforms [7,8]. Super-resolution experiments (stimulated emission depletion (STED) microscopy in combination with fluorescence correlation spectroscopy (FCS), STED-FCS) in HIV-infected cells proved that fluorescent analogues of cholesterol (labelled with Abberior STAR RED via a PEG-linker) and PIP_2_ (PIP(4,5)_2_ labelled with ATTO647N) were trapped at Gag assembly sites, as opposed to fluorescent analogues of sphingomyelin (SM, labelled with ATTO647N) and phosphoethanolamine (DPPE, labelled with ATTO647N or Abberior STAR RED) (Figure 1) [8]. Gag-induced lipid platforms were independently detected at the plasma membrane of transfected cells [7], which were enriched in fluorescent analogues of cholesterol, ganglioside GM1, SM (all TF-labelled) (Figure 1), and glycophosphatidylinositol (GPI) anchored proteins (labelled with enhanced green fluorescent protein (EGFP)). In the next assembly step, these platforms promote curvature-dependent protein sorting, which would ultimately lead to the unique lipidome and proteome of the viral envelope [7].

### 2.2. Imaging Lipids During HIV Entry

Many membrane-targeting compounds have been shown to inhibit — or promote — HIV-1 fusion, suggesting that the viral membrane constitutes a functional element of the entry process [16,17]. Still, the exact role of lipids during HIV-1 entry is unknown [38]. To the best of our knowledge, direct imaging of viral lipid species during membrane fusion has not been reported. For a detailed review on imaging HIV entry, uncoating, and nuclear entry, the reader could refer to the manuscript by Francis and Melikyan [39]. Instead, fluorescent imaging approaches have used the lipophilic dye DiD to localise HIV virions and resolve different entry steps [40,41,42,43,44,45,46]. When combined with a second intraviral marker (e.g., the Viral Protein R (Vpr) labelled with EGFP), the fluorescence signal from membrane dyes may be employed to report on viral envelope fusion with cell membranes [40,42]. This approach was exploited to distinguish endosomal from plasma membrane fusion of HIV [42]. In the case of plasma membrane entry, the membrane signal is diluted in the immense volume of cell membranes upon fusion, as opposed to the intraviral marker that stays trapped in the core and is finally released to the cytosol [41,42]. On the other hand, during endosomal entry, the membrane dye is not diluted due to the small volume of endosomes, but the intraviral marker signal disappears after capsid release to the cytosol [42,46]. More precisely, DiD fluorescence signal lost reports on hemifusion, i.e., the merger of the external leaflets of both membranes [43]. DiD was also employed to describe the co-localisation of the HIV-1 envelope with dynamin 2 (a protein involved in the secretory pathway) at viral entry sites [44]. 

An alternative approach to label the HIV membrane is the use of fluorescent versions of cell proteins that are incorporated into the envelope upon budding, such as ICAM-1 [47,48,49] or the N-terminal sequence of c-Src [50]. Fluorescent proteins may be straightforwardly engineered to report on specific environmental characteristics such as pH or membrane tension and voltage, as in the case of the ecliptic pHlourin-ICAM-1 (EcpH-ICAM) chimera, which bears a pH-sensitive mutant of the green fluorescent protein (GFP) [51]. Due to pH differences, EcpH-ICAM can be used to distinguish plasma membrane from endosomal viral entry [47,48,49]. Unfortunately, a certain bias was indicated for this pH sensor to destabilise the viral membrane [49]. Bias may be general to a label, entailing independent controls for activity and functionality.

### 2.3. Virus and Model Membranes

The properties of the HIV-1 membrane can also be studied on immobilised virions in the absence of producer or target cells. For these experiments, viruses are immobilised on a coverslip and then imaged. The main advantage of this approach is that a very high number of virions can be acquired at the same time thus, increasing statistical value and accuracy. This approach was initially employed to probe the colocalisation of viral fluorescent proteins and cell membranes, e.g., Vpr.GFP and DiD [40,41]. In more recent studies, immobilised virions have been imaged to quantify capsid release upon envelope permeabilization [52], where membrane staining was used as a reference (either DiD or EcpH-ICAM-1). 

Immobilised virions can also be employed to gain insights on the membrane biophysical properties. In a recent work, we made use of the polarity-sensitive dye Laurdan to quantify the molecular packing of the viral membrane of mature and immature immobilised HIV virions using confocal microscopy [53], similarly to previous in cuvette experiments [13]. Laurdan is a polarity-sensitive dye that changes its emission spectrum in response to solvent polarity. When incorporated into a lipid membrane, Laurdan reports on water accessibility to the membrane interior, which is determined by molecular order, i.e., lipid packing [54].

Still, fluorescence imaging of virions is generally limited by their small size below the resolution limit of conventional optical microscopes. Besides the above-mentioned super-resolution microscopy approaches, a remedy may be to extract lipids from HIV particles and reconstitute them in the form of GUVs. In a previous study, we employed this approach to study the membrane organisation of the HIV membrane and the effect of anti-viral compounds [14]. An alternative system to lipid extraction is the use of artificial model membranes, i.e., membranes made of simple lipid mixtures that resemble the chemical and biophysical properties of the HIV lipid envelope. Imaging of artificial model systems has been employed to determine lipid dynamics and molecular order of virus-like membranes [14,55,56].

## 3. Imaging the Envelope Glycoprotein

Imaging Env constitutes a challenging task. Adequate Env folding is very sensitive to changes in the amino acid sequence and thus, usual fluorescent protein-tagging strategies influence Env functionality and have thus so far shown limited success. Among the ca. 850 amino acids of Env, the most tolerant sequences to fluorescent protein insertion are the variable loops in gp120 (Figure 2, orange) [57]. Nakane et al. showed that GFP-opt, a derivative of superfolder GFP (Figure 2, green), can be inserted at the V4 and V5 loops while maintaining cell expression and functionality of Env in cells and virions (Figure 2) [57]. Moreover, the authors showed that this region is tolerant to the insertion of alternative fluorescent proteins such as mCherry. In another recent study, the V1/V2 variable loops were substituted by superfolder GFP, termed Env-isfGFP-ΔV1V2. Although Env-isfGFP-ΔV1V2 was unable to incorporate into virions, complementing with non-fluorescent Env-ΔV1V2 yielded fluorescent viral particles, which were significantly less infectious than virions packaging wild type Env [58]. Possibly, the main limitation to the introduction of fluorescent proteins in the Env sequence is their big size (Figure 2), which might interfere with multiple steps during entry, e.g., CD4 binding or gp41 refolding.

Antibody-labelling is another conventional strategy to label viral components. It has been broadly been employed in super-resolution microscopy studies of Env (as, for example, reviewed in [23]). The distribution of Env in assembly sites has been investigated by single molecule localisation microscopy (SMLM) to assess the role of the Env cytoplasmic domain and determine Gag-Env interactions [59,60], study the tetherin-mediated restriction of HIV release [61], and detect Env incorporation into virions [59,61]. Super-resolution STED microscopy and spectroscopy (e.g., STED-FCS, as already highlighted above) have been employed to study the maturation-dependant Env clustering [62] and mobility in single virions [53]. It must be noted that most common anti-Env antibodies origin from HIV-infected patients and neutralise Env activity, e.g., the broadly neutralising 2G12 antibody (Figure 2, blue). Thus, they can greatly affect the behaviour of Env by fixing specific structures. Introducing artificial antigen peptides in Env, such as the FLAG-tag [63], which are recognised by independent antibodies, can overcome this limitation. Anti-Env broadly neutralising antibodies themselves are also the subject of intensive research [64]. Their interaction with Env has been studied by means of STED microscopy [65] and FCS [66,67] in virions. Interestingly, some antibodies exert their neutralising activity through secondary interactions with viral lipids, which was recently studied by us using FCS on model membranes [68].

Enzymatically targeted incorporation of organic dyes constitutes an alternative bio-compatible Env-labelling strategy. Briefly, short peptide sequences are introduced in the target protein sequence. These peptides are the substrate of enzymes such as transglutaminases [69] or phosphopantetheinyl transferases [70], which catalyse bond formation between a specific amino acid within the sequence and an externally supplemented substrate, such as cadaverine or coenzyme A bound to a dye, respectively. This approach was exploited in a pioneering work where Munro et al. [71] measured the structural and conformational dynamics of native Env using single molecule Förster resonance energy transfer (FRET), monitoring fluorescence fluctuations due to distance changes between two nearby fluorescent labels (FRET label pair). Munro et al. could introduce a FRET label pair (Cy3B and Alexa Fluor 647, Figure 2) in the V1 and V4 loops of a single gp120 molecule per virus and resolve three distinct characteristic label pair distances, which corresponded to three Env conformational states [71,72,73]. Interestingly, this approach was also applicable to the study of the conformational dynamics of Influenza haemagglutinin [74] and Ebola GP glycoprotein [75].

Bio-orthogonal chemistry has also been exploited to label Env. It offers many advantages, such as high selectivity, biocompatibility and use of organic fluorescent dyes. Genetic code expansion to include non-canonical clickable amino acids at the V4 and V5 loops of Env, followed by click chemistry binding functionalised dye labels, showed specific labelling of Env at the plasma membrane of cells, although the changes in the nucleotide sequence decreased virus infectivity by one order of magnitude, probably due to lower Env incorporation efficiency [76]. Sugars of the Env glycoprotein (Figure 2, grey) can also be labelled by click chemistry [77]. Virions acquire clickable Env after the metabolic incorporation of clickable sugars into producer cells, which can subsequently be labelled with organic dyes such as Alexa Fluor 488. 

## 4. Future Directions

HIV infection is a complex process, which is not straightforward to be imaged. Most approaches discussed in this review are greatly limited by two aspects. First, labelling the envelope components, lipids or Env, certainly alters their behaviour, e.g., even the introduction of small organic dyes decreases viral titres. Second, virions can only host a limited number of fluorescent molecules due to their small size. Added to the fact that budding and fusion span for minutes, phototoxicity and photobleaching, i.e., light-induced changes or destruction in functionality and deprivation of the whole pool of labels per virion, respectively, become serious issues that need to be circumvented. 

Modern super-resolution techniques are still not perfectly suited to live cell imaging. SMLM requires acquisition times that are not always compatible with live measurements and STED microscopy may be greatly limited by photobleaching. In addition, imaging experiments may be recorded on apical cell membranes or ultimately in vivo and in tissue, i.e., the excitation and detection light beams have to travel through inhomogeneous sample regions with varying refractive indices. Such refractive index mismatch leads to distortions of the observation or focal spot (optical aberrations) and thus deteriorated image quality with respect to e.g., spatial resolution, signal-to-noise ratio and contrast, and may be compensated by improved or adaptive optics [81]. This is specifically important for super-resolution STED microscopy and STED–FCS recordings [82,83].

Another optimization step is the use of further fluorescence spectroscopy readouts such as fluorescence lifetime and anisotropy. Some commercial setups are readily implementing fluorescence lifetime imaging microscopy (FLIM) for spatially-resolved measurement of the fluorescence lifetime of fluorescent tags. Lifetime can provide additional information about the structure or environment of a fluorescent molecule and has successfully been employed in the HIV [44,46] and membrane biophysics [20] research fields. Not only established microscopy techniques are being refined, but also new ones are being developed. MINFLUX comes up as a promising solution, offering sub-nanometre localisation accuracy in live cells, while relying on low photon counts [84]. 

Advances in fluorescent dye design and synthesis are also providing new tools to unravel the mysteries of virus infection. With respect to new dyes, the reporting of unprecedented biophysical properties provides quantitative information about cell membranes [20], and has already been exploited in the study of HIV entry [46]. In the case of the small-sized HIV virions, exchangeable dyes constitute an interesting alternative to traditional membrane dyes, since they can circumvent photobleaching. Briefly, exchangeable membrane dyes transiently partition to lipid bilayers. Thus, the pool of fluorescent molecules at the membrane is renewed constantly as new dyes replace previously bleached ones [85,86].

In the future, sample preparation will surely evolve to increase the significance of microscopy-related findings. For example, most experiments described in this review article have been performed using the laboratory adapted strain NL4.3 (closely related to Env HXB2 sequence). This strain fails to represent most circulating isolates, presenting a lower entry efficiency and higher neutralisation susceptibility. Moreover, the majority of labelling strategies discussed above drastically decrease virus infectivity, e.g., insertion of fluorescent proteins in the Env sequence, click conjugation of dyes, or antibody staining. Future approaches will need to find minimally invasive labelling methods, for instance by identifying tolerant sites in the Env sequence. A detailed review on viral component labelling strategies was recently published by Sakin et al. [87]. Finally, most imaging studies have used artificial model systems, pseudotyped viruses, and immortal cell lines to study HIV infection. Upcoming studies will attempt to recapture the complex physiological context of HIV infection, e.g., performing experiments at 37 °C or using human primary lymphocytes.

Fluorescence microscopy of the HIV envelope is still a young field, but even if its most advanced approaches are far from being established, it has the potential to unravel many of the mysteries of HIV infection. We foresee that collaborative works between virologists, immunologists, microscopists, chemists, and biophysicist may ensure the success of this endeavour.

## Figures and Tables

**Figure 1 viruses-12-00348-f001:**
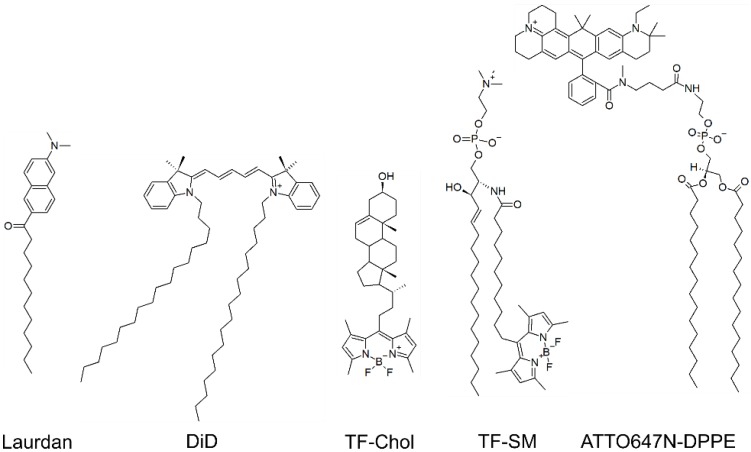
Examples of fluorescent molecules employed on HIV membrane research. DiD and Laurdan are membrane dyes. Fluorescent lipid analogues of cholesterol (labelled with TF, TF-Chol), sphingomyelin (labelled with TF, TF-SM) and DPPE (labelled with ATTO647N-DPPE) are also shown.

**Figure 2 viruses-12-00348-f002:**
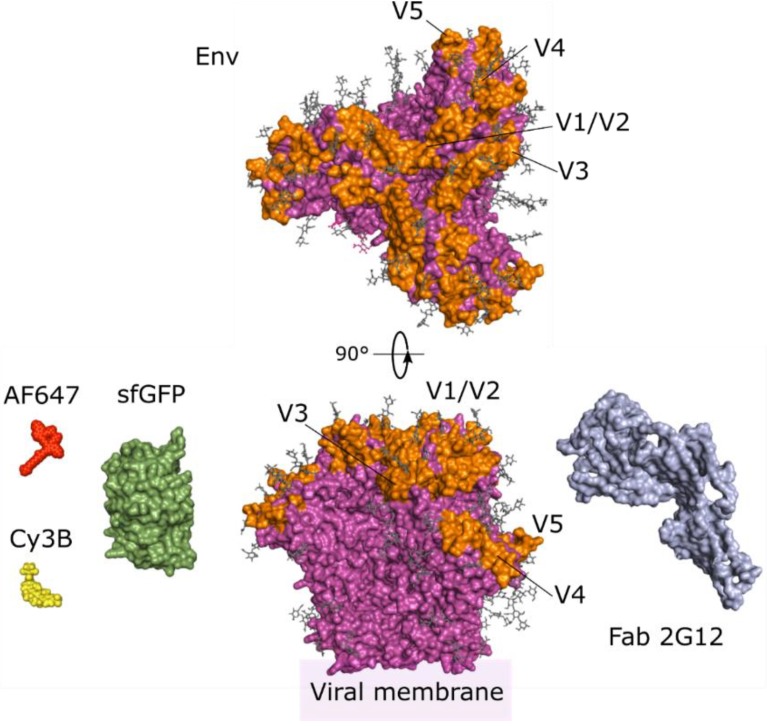
Structure of Env and previously employed labels. Env (magenta, PDB 5FUU [78]) is usually labelled in variable loops (V1-5, orange). Note that due to their disordered nature, a number of variable loop residues remain unresolved and are not depicted. Env glycans (grey sticks) can also be labelled by click chemistry [77]. Different labelling strategies have been employed to label Env, such as insertion of the fluorescent protein sfGFP (green, PDB 2B3P [79]) into variable loops, immunostaining by the 2G12 Fab (blue, PDB 6E5P [80]), or enzymatic incorporation of the organic dyes Cy3B (yellow) and Alexa Fluor 647-NHS ester (AF647, red). Note that up to three label units can be incorporated per Env protein.

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
