# Peer review of "Fluorescence Microscopy of the HIV-1 Envelope"

_viruses, 2020, doi:10.3390/v12030348_

Round 1

Reviewer 1 Report

This manuscript is a significant contribution to the field, and I recommend it to be accepted pending some suggested changes.

Comment# 1:

Line 90 – 93: For the sentence: “The lipid raft hypothesis proposed that ordered membrane platforms… denoted “lipid rafts” — for a detailed review on membrane organisation for example refer to [24] or [25].”

The quotes [24] or [25] are supposed to be used just as citations and cannot be used as being fully part of the sentence structure. My suggestion is to rewrite this text, and the long sentence could be divided in two and read: “The lipid raft hypothesis proposed that ordered membrane platforms…denoted “lipid rafts”. For a detailed review on membrane organisation refer to manuscripts such as those of Amaro et al [24] or Leventhal et al [25]”

Comment# 2:

Lines 131 – 132: We have the same issues than in comment # 1 in the sentence “…for a detailed review on imaging HIV entry, uncoating and nuclear entry refer to [34].

The citation [34] cannot be used as part of the sentence. It should be a reference independent to the sentence structure.

Comment# 3:

Lines 173- 174: For the sentence starting by “We employed this approach to study...”. This is most likely referring to a previous study conducted by the study team.

I would start the sentence by: “In a previous study, we employed this approach to study…”

Comment# 4:

Line 186 – 187: The sentence starting by “Moreover, the authors show that this region is tolerant...”

Please, put the sentence is the past tense. The sentence should read “Moreover, the authors showed that this region was tolerant…”

Comment# 5:

Line 215: The citation should come right after mentioning the authors name – and not at the end of the sentence. This could be rewritten as: “This approach was exploited in a pioneering work where Munro et al. [66] measured….”

Comment# 6:

Line 218: “The authors could introduce a FRET label pair…”; Please, specify who “The authors” is referring to.

Comment# 7:

Line 281-282: We have the same issue than in comment # 1 for the sentence: “A detailed review on viral component labelling strategies can be found in [81].”

Comment# 8:

Line 287: For the sentence “…approaches are far from being established, they have the potential….”; please clarify who or what “they” is referring to. Is this referring to “Fluorescence microscopy”? If that is the case, then “…they have the potential…” should be replace by “…it has the potential…”

Comment# 9:

Lines 288 – 289: The last sentence: “We foresee that collaborative works between virologists, immunologists, microscopists, chemists and biophysicist will ensure the success of this…” is coming out as a little bit too strong in predicting what will happen.

I believe that “…will ensure…” should be replace by “…may ensure…”

Author Response

Reviewer 1

This manuscript is a significant contribution to the field, and I recommend it to be accepted pending some suggested changes.

We thank the reviewer for their positive assessment and also for constructive suggestions.

Comment# 1: Line 90 – 93: (…) The quotes [24] or [25] are supposed to be used just as citations and cannot be used as being fully part of the sentence structure. My suggestion is to rewrite this text (…).

We have rewritten the text as suggested.

Comment# 2: Lines 131 – 132: (…) The citation [34] cannot be used as part of the sentence. It should be a reference independent to the sentence structure.

The citation is now independent of the sentence structure. (“(…) for a detailed review on imaging HIV entry, uncoating and nuclear entry the reader could refer to the manuscript by Francis and Melikyan [34]”)

Comment# 3: Lines 173- 174: (…) I would start the sentence by: “In a previous study, we employed this approach to study…”

We have adapted the text as suggested by the reviewer.

 Comment# 4: Line 186 – 187: (…) Please, put the sentence is the past tense (…).

The sentence has been corrected.

Comment# 5: Line 215: The citation should come right after mentioning the authors name (…).

We have adapted the text as suggested.

Comment# 6: Line 218 (…)Please, specify who “The authors” is referring to.

The text now reads “Munro et al could…” instead of “The authors could…”

Comment# 7: Line 281-282: We have the same issue than in comment # 1 (…).

The sentence is now independent from the citation.

Comment# 8: Line 287: (…) please clarify who or what “they” is referring to. (…) “…they have the potential…” should be replace by “…it has the potential…”

We have corrected this mistake. The text refers to “Fluorescence microscopy”.

Comment# 9: Lines 288 – 289: The last sentence (…) is coming out as a little bit too strong in predicting what will happen. I believe that “…will ensure…” should be replace by “…may ensure…”

We have replaced will by may as suggested by the reviewer.

Reviewer 2 Report

In this manuscript, Carravilla et al review fluorescence imaging studies of the envelope lipids and proteins, focusing on labelling strategies and model systems. The authors highlighted the advances in the field and the challenges of applying some the fluorescence imaging techniques in studying the structure, function and dynamics of the Env protein during virus assembly and upon infection. Some of the challenges are intrinsic to the labeling methods and others are related to the imaging technology. Overall, the manuscript is well written and is of interest to the general readers of Viruses. However, it can be improved further by addressing the minor issues listed below:

  1. Line 83-85, “The matrix (MA) domain binds… virion release”. Please provide references.
  2. Line 85-86, “MA binds the plasma membrane through… pocket and myristylation”. Please provide references that include Saad et al, PNAS, 2006, 103, 11364-9 and Vlach et al, PNAS, 110, 3525-3530.
  3. Line 99, “Nucleotide-Gag-membrane interactions…”. What do the authors mean by nucleotide? Do they mean RNA? There is no citation. Reference 28 has no RNA work.
  4. Line 222, “A recent paper on the application of smFRET to study the structure and dynamics of Ebola glycoprotein was just published in PloS Biol (Das et al, 2020, 18, e3000626).” Perhaps the authors need to cite it as well.     

Author Response

(…) Overall, the manuscript is well written and is of interest to the general readers of Viruses.

We thank the reviewer for their positive comments and for helpful suggestions.

1. Line 83-85, “The matrix (MA) domain binds… virion release”. Please provide references.

We have included references to the following articles:

-Fäcke, M.; Janetzko, A.; Shoeman, R.L.; Kräusslich, H.G. A large deletion in the matrix domain of the human immunodeficiency virus gag gene redirects virus particle assembly from the plasma membrane to the endoplasmic reticulum. J. Virol. 1993, 67, 4972–4980.

- Freed, E.O.; Orenstein, J.M.; Buckler-White, A.J.; Martin, M.A. Single amino acid changes in the human immunodeficiency virus type 1 matrix protein block virus particle production. J. Virol. 1994, 68, 5311–5320.

- Ono, A.; Ablan, S.D.; Lockett, S.J.; Nagashima, K.; Freed, E.O. Phosphatidylinositol (4,5) bisphosphate regulates HIV-1 Gag targeting to the plasma membrane. Proc. Natl. Acad. Sci. U. S. A. 2004, 101, 14889–14894.

- Saad, J.S.; Miller, J.; Tai, J.; Kim, A.; Ghanam, R.H.; Summers, M.F. Structural basis for targeting HIV-1 Gag proteins to the plasma membrane for virus assembly. Proc. Natl. Acad. Sci. U. S. A. 2006, 103, 11364–11369.

2. Line 85-86, “MA binds the plasma membrane through… pocket and myristylation”. Please provide references that include Saad et al, PNAS, 2006, 103, 11364-9 and Vlach et al, PNAS, 110, 3525-3530.

We have included those references and apologise for having missed such important contributions to the field.

3. Line 99, “Nucleotide-Gag-membrane interactions…”. What do the authors mean by nucleotide? Do they mean RNA? There is no citation. Reference 28 has no RNA work.

To avoid misleading the reader we have deleted the word nucleotide.

4. Line 222, “A recent paper on the application of smFRET to study the structure and dynamics of Ebola glycoprotein was just published in PloS Biol (Das et al, 2020, 18, e3000626).” Perhaps the authors need to cite it as well.

We have included a reference to this recent paper.